# Angiogenesis—An Emerging Role in Organ Fibrosis

**DOI:** 10.3390/ijms241814123

**Published:** 2023-09-15

**Authors:** Dan Wang, Ying Zhao, Yanni Zhou, Shaojie Yang, Xiong Xiao, Li Feng

**Affiliations:** Division of Liver Surgery, Department of General Surgery and Regeneration Medicine Research Center, West China Hospital, Sichuan University, Chengdu 610041, China; doc_wangsurgeon@163.com (D.W.); zhaoying_2021@163.com (Y.Z.); yannizhouscu@163.com (Y.Z.); sjyang0309scu@163.com (S.Y.); nessen2004@163.com (X.X.)

**Keywords:** lymphangiogenesis, fibrosis, VEGFC, VEGFR3, LYVE1

## Abstract

In recent years, the study of lymphangiogenesis and fibrotic diseases has made considerable achievements, and accumulating evidence indicates that lymphangiogenesis plays a key role in the process of fibrosis in various organs. Although the effects of lymphangiogenesis on fibrosis disease have not been conclusively determined due to different disease models and pathological stages of organ fibrosis, its importance in the development of fibrosis is unquestionable. Therefore, we expounded on the characteristics of lymphangiogenesis in fibrotic diseases from the effects of lymphangiogenesis on fibrosis, the source of lymphatic endothelial cells (LECs), the mechanism of fibrosis-related lymphangiogenesis, and the therapeutic effect of intervening lymphangiogenesis on fibrosis. We found that expansion of LECs or lymphatic networks occurs through original endothelial cell budding or macrophage differentiation into LECs, and the vascular endothelial growth factor C (VEGFC)/vascular endothelial growth factor receptor (VEGFR3) pathway is central in fibrosis-related lymphangiogenesis. Lymphatic vessel endothelial hyaluronan receptor 1 (LYVE1), as a receptor of LECs, is also involved in the regulation of lymphangiogenesis. Intervention with lymphangiogenesis improves fibrosis to some extent. In the complex organ fibrosis microenvironment, a variety of functional cells, inflammatory factors and chemokines synergistically or antagonistically form the complex network involved in fibrosis-related lymphangiogenesis and regulate the progression of fibrosis disease. Further clarifying the formation of a new fibrosis-related lymphangiogenesis network may potentially provide new strategies for the treatment of fibrosis disease.

## 1. Introduction

Organ lymphatic vessels can reabsorb tissue fluid back to the circulatory system to maintain tissue environmental homeostasis. In addition to tumors [1,2,3], organ transplantation [4], inflammation [5,6], and wound healing [7], the role of lymphangiogenesis in organ fibrosis has gradually attracted attention in recent years, such as in renal fibrosis [6,8,9,10,11], liver fibrosis [12,13], cardiac fibrosis [14,15,16,17], pulmonary fibrosis [18,19,20,21], tumor fibrosis [22] and peritoneal fibrosis [23,24]. Increasing evidence suggests that lymphangiogenesis is related to the onset and progression of fibrotic disease, and VEGFC/VEGFR3 is central in lymphangiogenesis. In addition, transforming growth factor-β (TGF) and hyaluronic acid (HA)/LYVE1 are also important factors in regulating fibrosis-related lymphangiogenesis. Lymphangiogenesis provides the delivery channel for inflammatory cells, which may be related to aggravating or alleviating fibrosis [25]. Currently, targeted lymphangiogenesis has become a potential strategy to treat fibrosis. However, it has not always achieved desirable results, and some have been ineffective again or even aggravated inflammation and fibrosis. Therefore, in the field of fibrosis-related lymphangiogenesis research, the following issues need to be clarified: (a) Is the effect of lymphangiogenesis on organ fibrosis beneficial or harmful, and does it differ in different organs? Are there differences among the different stages of fibrosis? (b) How does lymphangiogenesis influence the progression of organ fibrosis? (c) What are the core control factors or networks of lymphangiogenesis? (e) How does the complex interplay between lymphangiogenesis and the immune response regulate fibrosis development? Therefore, based on the existing fibrosis data, we summarized the role of lymphangiogenesis in fibrosis, including the source of LECs, the regulatory mechanism, and fibrosis therapeutic strategies targeting lymphangiogenesis in organ fibrosis.

## 2. Lymphangiogenesis Pathway

The lymphatic system consists of capillary lymphatic vessels, precollecting lymphatic vessels and collecting lymphatic vessels [26,27]. Lymphatic capillaries consist of a single layer of lymphatic endothelial cells, which do not cover the “lymphomuscular” cells, showing discontinuous “button” connections. This special linkage allows fluid, antigens, and immune cells to enter the lymphatic capillaries. The anterior and collecting lymphatic vessels are covered with lymphomyocytes and are located downstream of the capillaries in a “zipper” connection. It has been reported that capillary vessels lose this “button-on-like” structure and take on a less permeable “zipper-like” structure [3,28]. Changes in the connective structure may impair the transport of fluids and substances to the lymphatic capillaries, thereby reducing their clearance from tissues.

Classical lymphangiogenesis refers to the formation of germination from embryos and differentiation from endothelial cells of venous vessels in adults. Lineage tracking revealed that LECs were not a single source but were heterogeneous in origin in different organs, which may be related to differences in lymphatic vessel function [28]. Lymphoprogenitor cells, the source of endothelial cells in lymphatic vessels, are mainly hematopoietic stem cells [29], mesenchymal stem cells [30], fat-derived stem cells [31,32], and bone marrow stem cells [33]. Lymphangiogenesis refers to an increase in the number of lymphatic vessels or the enlargement of a network of lymphatic vessels associated with an increase in the area of lymphatic vessels, including the following pathways: (a) The original lymphatic vessels form new lymphatic vessels by budding, and the VEGFC/VEGFR3 signal is an important stimulus factor for the continuous budding of lymphatic vessels [34,35]. (b) Peripheral lymphatic progenitor stem cells can migrate to lymphangiogenesis sites and differentiate into lymphatic endothelial cells, which also requires VEGFC. (c) Macrophages can transdifferentiate into LECs under VEGFC and other factors [10,36].

The common lymphangiogenic pathway considers VEGFC/VEGFR3 and PROX 1 signaling as the main molecular drivers that play a crucial role in both physiological and pathological conditions [37]. PROX1 activates the expression of VEGFR3, and VEGFC-mediated signals can maintain PROX1 levels, forming a positive feedback loop supporting the development of lymphatic vessels [38]. In addition, ADAMTS3 and CCBE 1, as well as ADAMTS2/ADAMTS14, process and activate pro-VEGFC at different stages, forming mature VEGFC [39,40]. Multiple signaling mediates lymphangiogenesis, as well as crosstalk with VEGFR3-PROX1. VEGFR3 activation leads to the phosphorylation of protein kinase B (PKB)/AKT, which promotes LEC migration, proliferation, and survival [41]. Exploring the molecular signaling mechanisms of lymphangiogenesis may provide new ideas on disease pathogenesis and prognosis (Figure 1).

Fatty acid β-oxidation (FAO) is involved in lymphangiogenesis, and acetyl-coenzyme A (ac-CoA) is central in the FAO-mediated lymphangiogenesis, which upregulates the expression of PROX1 [27,42]. PROX1 can promote CPT1A expression, an FAO rate-limiting enzyme, accelerating the FAO process to produce more ac-CoA and forming the positive regulation of PROX1/ac-CoA [42]. In addition, the lipid droplets (LD) produced by LECs themselves through autophagy can also supply FAO as a raw material [43].

In an inflammatory environment, LECs can express multiple chemokines, such as CCL21, attracting CCR7+ immune cells to infiltrate into their periphery, including macrophages and T cells [2]. Meanwhile, (TLR4)-NF-κB signaling was enhanced in LECs, upregulating lymphangiogenesis by increasing the expression of PROX1 and VEGFR3 [44,45]. Recruited macrophages can participate in the remodeling of lymphatic vessels. TLR4 signaling in macrophages enhances the expression of VEGFC and VEGFD and promotes lymphangiogenesis. However, VEGFC/VEGFR3 signaling activates P13K-Akt signaling, promotes SOCS 1 expression and suppresses (TLR4)-NF-kB signaling, thus attenuating the release of inflammatory factors as well as the inflammatory response [46]. In addition, in the subsequent descriptions, we also mentioned that macrophages can partially transdifferentiate into LECs to promote lymphangiogenesis.

In addition to VEGFR3 and PROX1, there are several other lymphangiogenesis-related signals reported sporadically. Betterman et al. showed that FAT4 is required for lymphatic vessel morphogenesis throughout development and that FAT4 mutations can cause Hennekam syndrome, characterized by dilated lymphatic vessels, lymphedema, facial abnormalities, and growth retardation [47]. Indeed, FAT4, CCBE1, and ADAMTS3 mutations underlie Hennecham syndrome, which suggests that these three genes may function in a common pathway [47,48]. As mentioned earlier, these several genes are also involved in the maturation of VEGFC, which may be a potential mechanism involved in the abnormal development of lymphatic vessels in Hennecham syndrome. Sox 18 directly activates PROX1 transcription by binding to its proximal promoter. Overexpression of Sox18 induces overexpression of PROX1 and other lymphatic endothelial markers, whereas Sox 18-null embryos show completely blocked differentiation of lymphatic endothelial cells from the vein [49]. 

## 3. The Role of Lymphatic Vessels in Tissue and Organs

Different from the core vascular system of the heart, the lymphatic system is a one-way, open transport network without a central driving force, and is an important defense system of the body that collects tissue fluid, cells, and proteins and filters them into specialized blind lymphatic capillaries and maintains the homeostasis of the interstitial fluid [50]. At the same time, after passing through secondary lymph nodes rich in infantile lymphocytes, foreign antigens are detected to trigger an adaptive immune response and play the role of immune surveillance. Second, lymphatic vessels can act as transport channels for immune cells and have bi-inflammatory regulatory effects of excreting inflammatory infiltrating cells and producing inflammatory cytochemokines to maintain the immune response. In addition, lymphatic vessels are involved in the absorption of fatty acids and fat-soluble vitamins from the gut [50,51]. With the continuous emergence of new functions of lymphatic vessels, it has been found that lymphatic vessels have significant plasticity and heterogeneity and can actively participate in the regulation of the microenvironment of multiple organs and tissues.

### 3.1. Kidney

The renal lymphatic system is important for the humoral balance in the kidney, as well as for the transport of immune cells and antigens. Lymphoid dysfunction may cause interstitial edema in the kidneys, and increased abnormal lymphatic vessels can exacerbate the inflammatory infiltrates. Changes in lymphangiogenesis were observed in kidney diseases. In human and mice kidney transplantation, upregulated lymphangiogenesis was associated with lymphocyte infiltration, allograft fibrosis and impaired graft function, and therapies that inhibit lymphangiogenesis improved the outcome and survival of allografts [52,53,54,55]. In addition, in multiple renal disease biopsies, from moderate chronic kidney disease to end-stage kidney disease, increased lymphangiogenesis and lumen cross-sectional areas were found [56]. The replicated finding in animal models of kidney disease was also observed, but there was a coincident conclusion about lymphangiogenesis on nephropathy; detailed information can be found regarding organ fibrosis [57,58]. As with other organs, VEGFC/VEGFD -VEGFR3 mediates the lymphangiogenesis of the kidney. Notably, only in the case of renal injury is VEGFC/D expression up-regulated by macrophages, renal tubular cells, and podocytes, and glomerular endothelial cells express VEGFR3/2 [57,59].

### 3.2. Liver

The liver is recognized as the largest lymph-producing organ. Approximately 25% to 50% of the lymph fluid flowing into the thoracic duct is produced in the liver. Lymphatic formation is greatly affected by hemodynamic changes in the microcirculation in the liver [12,60,61]. Hepatic lymphocytes are formed by the filtration of plasma components through sinusoidal endothelial cells into the Disse space (the space between sinusoidal endothelial cells and hepatocytes). Although the detailed lymphoid components of the liver are not well defined, these components may include cellular byproducts excreted by hepatocytes in the perinusional space [62]. The lymph drains into lymphatic vessels located in three areas: the portal vein, the hepatic vein, and the subcapsular area, and drains into the hilar lymph nodes of the liver. Lymphatic vessels in the portal vein are the main parts of liver lymphatic drainage, accounting for approximately 80% of the lymph produced by the liver [62]. However, the lymphatic system of the liver is rarely mentioned, and few studies have reported its role in pathological situations. In cirrhosis, massive fibrosis leads to structural deformation, increased blood flow resistance and hydrostatic pressure in hepatic sinuses, increased filtration of plasma through hepatic sinuses, and significantly increased lymph production [63]. Many malignant tumors secrete lymphangiogenic factors, such as VEGFC and VEGFD, and promote lymphangiogenesis in adjacent tissues, thereby helping tumor cells metastasize through lymph nodes. The expression of VEGFC in hepatocellular carcinoma is positively correlated with tumor size and number but negatively correlated with disease remission rate and disease-free survival time [38]. Therefore, blocking VEGFC-mediated lymphangiogenesis may be a potential therapeutic strategy. Currently, no selective drugs that specifically inhibit lymphangiogenesis have been approved for clinical use. Phase I trials of the anti-VEGFR3 monoclonal antibody LY3022856/IMC-3C5 in patients with advanced solid tumors showed minimal antitumor effects [64]. Lymphangiogenesis plays a beneficial role in reducing inflammation during the early stages of liver transplantation [26]. In patients with nonalcoholic fatty liver disease, increased lymphatic vessel density was observed in areas of fibrosis and immune cell infiltration. Lymphatic transport activity was impaired in mice with nonalcoholic steatohepatitis, but this was avoided by administration of recombinant vascular endothelial growth factor C (rVEGFC) while improving liver inflammation, suggesting that drugs with lymphangiotropic properties may be a new treatment strategy for nonalcoholic steatohepatitis [13].

### 3.3. Heart

The mammalian heart has large and complex lymphatic vessels that maintain fluid balance in cardiac tissue, protect the heart from infection, prevent myocardial edema, and restore cardiac function after infarction, involving various cardiovascular diseases [65,66]. In human and mouse atherosclerotic plaques, lymphatic vessels and increased lymphatic vessel density in the adventitia coincide with lesion severity [67,68], and VEGFC-induced lymphangiogenesis alleviated the cholesterol in the plaque [69]. Similarly, in myocardial infarction (MI), VEGFC secreted by macrophage-driven lymphangiogenesis promotes cardiac infarct healing as well as tissue repair [70]. Blockade of VEGFC signaling by soluble VEGFR3 (sVEGFR3) leads to impaired cardiac lymphoid morphology and increased mortality due to myocardial infarction in mice [71]. However, in contrast to the above findings, in heart transplantation, it is thought that cardiac lymphocangiogenesis is related to chronic rejection in allograft recipients, and blocking immune cell influx due to lymphangiogenesis enhances graft survival [72]. Moreover, in recent years, studies have found that the zebrafish has an amazing ability to repair the heart muscle, with many new lymphatic vessels covering the damaged area, and blocking lymphangiogenesis, the zebrafish cannot repair the myocardium [73,74]. This suggests that lymphangiogenesis may play a very important role in cardiac repair [73,74]. In addition, pro-lymphangiogenic treatment performed in mice has been shown to promote post-MI heart healing by reducing fluid retention and improving inflammatory cell clearance [70,75]. Therefore, lymphatic vessels appear to play different roles in different disease states.

### 3.4. Other Organs

In addition to the kidney, heart, and liver, lymphangiogenesis has also been studied in other organs, such as the gut and brain. The lymphatic vessels of intestinal tissue have important functions in fat absorption, maintaining intestinal immune function and promoting intestinal homeostasis [76]. The brain parenchyma has no lymphatic vessel structure, but the removal of cellular debris and waste products in the central nervous system is attributed to the lymphatic system [77]. In recent years, it has been believed that meningeal lymphatic vessels (mLVs) are the potential drainage circuit of macromolecular clearance, and also the way for immune cells to travel from the cerebrospinal fluid into the peripheral lymphatic system [78]. The link between mLV dysfunction and impaired brain waste clearance is related to the diagnosis and treatment of various diseases [79]. Moreover, VEGFC/VEGFR3 induced lymphangiogenesis in multiple organs, so defining the source of VEGFC in different situations is necessary for the role of lymphatics in organ physiology and pathology state [80].

## 4. Lymphangiogenesis and Fibrotic Disease

Lymphangiogenesis has been observed in the course of multiple organ fibrosis studies, including renal fibrosis, cardiac fibrosis, pulmonary fibrosis, peritoneal fibrosis, liver fibrosis, tumor fibrosis, and so on (Table 1). These studies confirmed the important role of lymphangiogenesis in the pathological process of fibrosis, potentially indicating that lymphangiogenesis is related to the progression or inhibition of fibrotic disease. Although lymphangiogenesis was found in fibrotic disease, the purpose of lymphangiogenesis and its role in fibrotic disease are not completely clear, and the current findings are not uniform.

**Table 1 ijms-24-14123-t001:** Lymphangiogenesis in fibrotic diseases.

Organ Fibrosis	Effect	Role of Lymphangiogenesis in Fibrosis	Lymphangiogenesis Mechanism	Reference
Kidney	Bad	Increase inflammatory responses	VEGFC/VEGFR3-dependent autophagy and polarization of macrophages promote the lymphangiogenesis	[8]
Kidney	Bad	Promoting intrarenal inflammation by CCL21/CCR7	Lymphangiogenesis-related factors	[5]
Kidney	Good	Ameliorating inflammation and fibrosis	VEGFC/VEGFR3 induce lymphangiogenesis	[81]
Kidney	Bad	Propagating an inflammatory feedback loop, aggravating inflammation and fibrosis	CD137-CD137L induce the autophagy of LECs to promote the lymphangiogenesis	[82]
Kidney	Bad	-	CTGF-VEGFC pathway	[83]
Kidney	Related	-	VEGFC/VEGFD promote lymphangiogenesis	[84]
Kidney	Unrelated	Tubulointerstitial fibrosis and inflammation are separate from lymphangiogenesis	VEGFC/VEGFR3 pathway	[85]
Heart	Good	Draining the interstitial fluid to reduce cardiac edema and collagen synthesis	VEGFC produced by M2b macrophages promote the lymphangiogenesis	[86]
Heart	Bad	LECs can be partially converted to phenotypically altered myofibroblast-like cells	Aldosterone triggers inflammatory response leading to increased VEGFC secretion	[87]
Heart	Good	Transport of immune cells to draining mediastinal lymph nodes	VEGFC stimulates cardiac lymphangiogenesis	[88]
Heart	Good	Reducing heart inflammation and fibrosis	VEGFC accelerates cardiac lymphangiogenesis	[14]
Heart	Good	Reduce macrophages, B cells, and perivascular fibrosis	-	[89]
Lung	Bad	LECs transdifferentiate into myofibroblasts	VEGFC pathway	[90]
Lung	Good	Lymphatic vessels are involved in fibrotic maturation mainly through excessive protein and fluid drainage	Hyaluronic acid and activated CD11B + macrophage partially drive the lymphangiogenesis	[18]
Peritoneum	Bad	Inhibition of lymphangiogenesis alleviated worsening net ultrafiltration	VEGFD promotes lymphangiogenesis	[23]
Peritoneum	Bad		TGFβ promotes VEGFC production and lymphangiogenesis	[24]

In fibrotic disease, the number of lymphatic vessels changes, but it is not clearly harmful or beneficial to the disease, showing only some correlation with fibrosis. El-chemaly et al. found that the lymphatic vessel diameter was correlated with the severity of idiopathic pulmonary fibrosis and that lymphangiogenesis is involved in remodeling of the alveolar microenvironment [18]. In addition, Ishikawa et al. showed a correlation between lymphangiogenesis and cardiac fibrosis stage through 40 human necropsy heart tissues [91]. In animal model studies, Lee et al. observed an increased number of LYVE1+ lymphatic vessels in the mouse unilateral ureteral obstruction (UUO) model, which was associated with the presentation of renal fibrosis [84]. Similarly, Yazdani et al. induced lymphangiogenesis in a rat renal proteinuria model by doxorubicin [85]. In addition, lymphatic fluid production increases greatly in the liver during cirrhosis and fibrosis, accompanied by a large amount of lymphangiogenesis, which then triggers the immune response in the liver and participates in the fibrosis process. The process of lymphangiogenesis may be related to portal hypertension [12,13]. However, in the acute liver injury model, lymphangiogenesis seemed to contribute to recovery from liver injury [92]. In addition, ascites formation because of increased lymphatic fluid production is a recognized clinical manifestation of hepatic lymphatic system disorder in patients with liver fibrosis and cirrhosis [26].

Lymphatic vessels, as immune cell transport channels, will rapidly recruit immune cells after local injury stimulation, promoting the course of fibrosis. In renal fibrosis, the lymphangiogenesis-activated inflammatory response promotes the progression of renal fibrosis, and hyperplasia LECs of the kidney and the corresponding draining lymph node upregulate the chemokine CCL21, induce infiltration of a large number of CCR7 receptor-expressing lymphocytes, macrophages and dendritic cells into the kidney and draining lymph nodes, and expand the inflammatory response and promote fibrosis [5]. Similarly, Zhang et al. and Wei et al. found that lymphangiogenesis promoted the inflammatory response and accelerated fibrosis progression in the UUO model [8,82]. Clinical dilated cardiomyopathy samples had increased lymphatic vessel density despite reduced lymphatic vessel size. In a model of transverse aortic constriction (TAC)-induced cardiac fibrosis, significant lymphangiogenesis was associated with reduced levels of cardiac macrophages, B cells, and perivascular fibrosis [89]. However, no significant lymphangiogenesis was observed, but the inhibition of lymphangiogenesis increased perivascular fibrosis and accelerated the development of left ventricular dilatation and dysfunction [89]. In addition, LEC transdifferentiation into myofibroblasts is also the cause of lymphangiogenesis to promote fibrosis. In a rat lung and heart fibrosis model, lymphatic endothelial cells were observed to coexpress α-SMA and β, namely, new lymphatic endothelial cells can transdifferentiate into myofibroblasts or undergo the endothelial mesenchymal transition process, and a large deposition of collagen I was observed around the lymphatic vessels [87,90]. Myofibroblasts are considered central factors during tissue healing and tissue fibrosis. In the process of normal wound repair, myofibroblasts exist briefly, prompting wound contraction and connective tissue repair. However, in fibrotic tissue, myofibroblasts persist, excessive extracellular matrix deposits occur, the tissue structure is reshaped, and the normal tissue structure is destroyed, which can eventually lead to dysfunction and even failure. Moreover, lymphangiogenesis is also involved in peritoneal fibrosis, and proper fluid balance is important for good clinical outcome and survival of peritoneal dialysis patients. Lymphangiogenesis worsened net ultrafiltration in the methylglyoxal (MGO)-induced peritoneal fibrosis model [23,24].

In contrast, it was also reported that lymphangiogenesis alleviates the course of fibrosis, benefiting from its function of clearance of immune cells in the injured microenvironment. VEGFC/VEGFR3-induced lymphangiogenesis protects against myocardial injury and suppresses the progression of cardiac hypertrophy to cardiac failure [93]. Hasegawa et al. induced lymphangiogenesis in UUO model mice by injecting recombinant human VEGFC with an osmotic pump, significantly reducing macrophage infiltration, inflammatory cytokines and TGFβ levels, and alleviating renal fibrosis [81]. Injection of M2b macrophages into a rat model of cardiac fibrosis also promoted lymphangiogenesis, alleviated cardiac fibrosis and improved cardiac function [86]. Similarly, Vieira et al. found that lymphangiogenesis increased the clearance of immune cells after MI and drained cardiac immune cells to the mediastinal lymph nodes to relieve the inflammatory response after cardiac injury [88]. In an angiotensin-induced model of cardiac inflammation and fibrosis, causing cardiac dysfunction and impaired lymphatic transport at 6 weeks, VEGFCc156s improved cardiac fibrosis by enhancing lymphangiogenesis [15]. Poor lymphatic vascular remodeling after myocardial infarction leads to the reduced lymphatic transport capacity of the heart, causing myocardial edema, which improves the myocardial fluid balance and reduces cardiac inflammation, fibrosis and dysfunction [14]. Lymphangiogenesis peaked at 14–28 days in a bleomycin-induced lung injury model, accompanied by expansion of CCL 21 in lung lymphoid tissue and then decreased at 56 days with resolution of lung injury. Transgenic overexpression of VEGFC expanded the lung lymphatic network, reduced macrophage accumulation and fibrosis, and accelerated recovery after bleomycin treatment [21]. Shenone IIA sodium sulfonate (sodium tanshinone IIA sulate, STS) promotes lymphangiogenesis in silicosis rats and relieves inflammation, oxidative damage and fibrosis through the VEGFR3/PI3K/AKT signaling pathway [94]. In conclusion, lymphangiogenesis has a comprehensive protective role in lung injury and fibrosis.

Although the association between lymphangiogenesis and fibrosis has been confirmed by comparable studies, some studies have shown that lymphangiogenesis is not associated with fibrosis and that intervention with lymphangiogenesis does not improve fibrosis. Yazdani et al. inhibited lymphangiogenesis in rats with albuminuria by VEGFR3 blockers but had no significant effect on markers of inflammation and fibrosis or albuminuria [85]. In a proteinuria model, lymphangiogenesis precedes the progression of fibrosis in the 6th week, without myofibroblast and macrophage infiltration, and the number of myofibroblasts increases significantly in the 12th week, while collagen deposition and macrophages do not increase [95]. Therefore, the relationship between lymphangiogenesis and fibrotic diseases is different in different diseases, which may be related to the organ heterogeneity of LECs.

## 5. The Origin of LECs in Fibrosis-Lymphangiogenesis

It has long been assumed that lymphatic vessels are derived from early veins and that lymphatic structures occur through the sprouting of lymphatic vessels. However, recent evidence indicates that lymphatic vessels can also be differentiated from cells from other pathways [96]. It has been established that lymphangiogenesis is associated with organ fibrosis. With regard to the origin of LECs in fibrosis, some studies have shown that they come from original lymphatic endothelial cells or from bone-marrow-derived macrophages. Pei et al. explored the origin of LECs in renal fibrosis using a “symbiotic model” that made symbiotic surgical connections between wild-type mice and transgenic mice with a green fluorescent protein (GFP) and a “BM chimera” that was generated by transplanting GFP strain bone marrow cells into wild-type mice. However, only low levels of GFP colocalized with LYVE1, suggesting that macrophages are only a small part of the source of LECs [5]. D2-40 costaining with PCNA demonstrated that the source of lymphatic endothelial cells during lymphangiogenesis in the kidney and lymph nodes was mainly the result of preexisting lymphatic endothelial proliferation rather than direct transdifferentiation of bone-marrow-derived cells [5]. Similarly, Zhang et al. obtained similar results through a mouse UUO model and cell experiments, which showed that bone-marrow-derived macrophages are involved in the formation of lymphatic vessels and that M1-type macrophages account for the major proportion of transdifferentiation. VEGFC, as a core factor, promotes transdifferentiation into lymphatic endothelial cells by inhibiting autophagy in M1 macrophages. At the same time, with the progression of renal fibrosis, the infiltration of M2 macrophages with a high degree of autophagy increased, while the infiltration of M1 macrophages gradually decreased [8]. Lymphangiogenesis was reduced after macrophage clearance in a mouse UUO model, which further provided support for the correlation between macrophages and lymphangiogenesis [84]. However, Yazdani et al. did not achieve the same results, and clearance of macrophages by chlorophosphonate liposomes failed to significantly inhibit lymphangiogenesis and fibrosis progression [85]. In addition, lymphangiogenesis occurs in the recipient after kidney transplantation and lymphoid progenitors may arise from circulating macrophages and eventually integrate into the growing lymphatics [52]. El-Chemaly et al. isolated CD11b macrophages from patients with idiopathic pulmonary fibrosis. In in vitro cultures, CD11b macrophages could form a tube structure, while CD11B macrophages in healthy controls could not [18]. Maruyama et al. demonstrated in a mouse keratitis model that CD11b+ macrophages alone can form lymphatic tubular structures expressing the lymphatic endothelial markers LYVE-1 and podoplanin [97].

## 6. Modulator of Fibrosis-Related Lymphangiogenesis

Based on the changes and effects of lymphangiogenesis in multiple organ fibrosis, it is particularly important to clarify its regulatory mechanism. At present, VEGFC/VEGFR3, VEGFA, VEGFD, FGF2 and other factors are considered to be related to fibrosis lymphangiogenesis. According to the literature reports, we have made a mechanistic diagram of fibrosis lymphangiogenesis (Figure 2).

### 6.1. VEGFC/VEGFR3

As a regulatory pathway of lymphangiogenesis, VE GF C/VEGFR3 plays an important role in the growth and remodeling processes of lymphatic vessels [98,99]. In addition to mediating tumor lymphangiogenesis [100,101,102], it has been implicated in multiple fibrosis studies, increasing the activity of lymphatic endothelial cells and the expression of adhesion molecules and promoting the proliferation, migration, and tube formation of lymphatic endothelial cells [81,84,103,104]. In the UUO mouse fibrosis model, a VEGFC-specific short hairpin RNA lentiviral vector was transfused back into immunodeficient mice after knockdown of macrophage VEGFC, which significantly decreased Lyve1-positive lymphatic vessels, suggesting that VEGFC secretion from macrophages plays an important role in lymphatic angiogenesis. Moreover, M2 macrophages secreted more VEGFC than M1 macrophages [84]. Additionally, in a mouse UUO model, VEGFC promotes lymphangiogenesis by inhibiting the autophagy level of M1 macrophages and leading to their differentiation into lymphatic endothelial cells [8]. VEGFC can promote macrophages to secrete CD137L, act on the lymphatic endothelial cell receptor CD137, promote autophagy through the PI3K/AKT/mTOR pathway, induce the transition from LC3-I to LC3-II, and enhance the migration and proliferation ability of lymphatic endothelial cells [82]. In myocardial infarction models, VEGFC secreted by proinflammatory macrophages drives lymphangiogenesis and extensive remodeling of the cardiac lymphatic network, maintaining immune cell homeostasis and effective tissue repair during postinfarction healing [93]. In addition, downregulation of VEGFR3 can impair cardiac lymphangiogenesis, and chronic stress overload can lead to cardiac edema, resulting in cardiac dysfunction [93]. sVEGFR3 transgenic mice and Chy mice, which have an inactivating mutation in the VEGFR3 gene, confirmed that VEGFR3 downregulation altered the structure of the cardiac lymphatic network, resulting in increased vascular leakage and mortality after MI [71]. In a model of fibrosis induced by temporary occlusion of the left coronary artery, although robust intracyocardial capillary lymphangiogenesis was induced, poor remodeling of the epicardial precollection led to myocardial edema, and exogenous infusion of VEGFC accelerated cardiac lymphangiogenesis and slowed edema and fibrosis. In acute liver injury, VEGF C/VEGFR3 signaling activation also promotes lymphangiogenesis, and exogenous VEGF C can regulate the polarization of Kupffer cells to LECs and alleviate the liver injury caused by ischemia and reperfusion. Kupffer cells can also produce VEGFC and amplify the VEGF C/VEGFR3 signaling pathway, so lymphangiogenesis may also play a key role in liver fibrosis [92,105,106,107,108]. However, the mechanism by which autophagy induces Kupffer cells into the M1 or M2 type and then induces transdifferentiation into LECs during liver fibrosis is not clear.

### 6.2. TGFβ

TGFβ is a classical profibrotic factor and an inhibitor of lymphangiogenesis. TGFβ inhibited the expression of prox-1 and LYVE 1, as well as proliferation, migration, and tube formation in human LECs in vitro [25,109]. In the mouse UUO model, TGFβ and VE GF C were upregulated simultaneously, and although TGFβ induced VE GF C expression in vitro and in vivo, TGFβ inhibited lymphangiogenesis mediated by the VEGFC/VEGFR3 pathway [84,110]. Interestingly, VEGFD can rescue the process of VE GFC inhibition by TGFβ [84]. Connective tissue growth factor (CTGF) is an important determinant of fibrotic tissue remodeling, which is related to the occurrence and progression of renal fibrosis [111,112]. TGFβ induces CTGF expression in renal tubular cells, and CTGF can also regulate the expression of TGFβ through direct physical action [113]. In a mouse model of renal fibrosis and ischemia reperfusion, CTGF knockdown resulted in a significant reduction in VEGFC and lymphatics, suggesting that CTGF contributes to VEGFC secretion and lymphangiogenesis [83]. Similar to the effect of TGFβ on VEGFC, full-length GTGF, but nonfragmented GTGF (NH2-terminal fragment of CTGF or COOH-terminal fragment of CTGF), induced VEGFC production in renal tubular cells through direct binding to VEGFC but inhibited VEGFC-mediated growth of LECs [83].

### 6.3. Hyaluronic Acid (HA)/LYVE1

Lymphatic endothelial hyaluronic acid receptor 1 (LYVE 1) is a homolog of CD44, the main cell surface receptor of hyaluronic acid (HA) in LECs, and an important factor in lymphangiogenesis [114,115]. Hyaluronan (HA), a large nonsulfated glycosaminoglycan in the extracellular matrix with a degradation fragment known as low molecular weight hyaluronic acid (LMW-HA), has been reported as an important regulator of angiogenesis [116]. Myofibrotic cells produce a large amount of ECM, which contains HA, due to the special structure of lymphatic vessels that allow HA to bind to lymphatic endothelial cells [25]. HA promotes the proliferation, migration, and tube formation of LECs by binding to LYVE 1, a receptor located in LECs [117]. In human idiopathic pulmonary fibrosis, HA and CD11b macrophages in the alveoli contribute to lymphangiogenesis, but the specific mechanism is not addressed [18]. In the mouse UUO model, HA accumulation in the cortical stroma was positively correlated with lymphatic vessel number, and TGF-β1 stimulated increased hyaluronan synthase (HAS) mRNA expression and HA production in bone marrow macrophages, transferring mHAS2 and mHAS3 knockdown CD11b macrophages to SCID mice and partially reducing fibrosis-induced lymphangiogenesis [116]. Elimination of macrophages with clodronate reduced fibrosis-induced HA accumulation and lymphangiogenesis, and macrophages may be one of the sources of HA [116].

### 6.4. MicroRNA

In addition, some reports have documented the involvement of miRNAs in the biological processes of LECs. miRNAs are small endogenous non-coding RNA with a length of about 22 nucleotides that play an active role in a variety of pathophysiological mechanisms by regulating a variety of cellular processes, including cell proliferation, differentiation, apoptosis, invasion, and migration [118]. At present, there are few studies on the relationship between miRNA-LECs and organ fibrosis, and most studies focus on the effect on the tumor microenvironment (TME). Azzarito et al. found that MCF-7 cells can intervene in lymphatic growth through endocrine action, and miR193a-3p can promote breast cancer by inhibiting LEC’s biological behaviors [119,120]. The authors also performed microarray analysis to explore the potential mechanisms involved in the inhibition of miR193a-3p on LECs; the enriched pathways were cell adhesion, apoptosis, ABC transporter, IL-17 signaling pathway, and necroptosis [119]. In cervical cancer, miR-1468-5p promoted the upregulation of lymphatic PD-L1 and lymphangiogenesis, thereby impairing T-cell immunity [121]. In addition, high levels of miR-1468-5p in serum were associated with TME immunosuppression and poor prognosis in patients with cervical cancer [121]. The differential biological behavior of tumors caused by different lymphatic vessel states mediated by miRNA is due to the unclear role of lymphatic vessels in tumors.

## 7. Prospective Therapeutic Strategy for Fibrosis Targeting Lymphangiogenesis

The regulation of lymphangiogenesis has gradually become a new platform for studying therapeutic strategies in fibrotic diseases. Overall, the selective regulation of fibrosis-related lymphangiogenesis should focus on addressing the inflammatory response. To date, VEGFC/VEGFR3 is considered to be the most important and promising candidate for lymphangiogenesis therapy in this field, and other major targets are LYVE 1 and macrophages. However, due to the differences in fibrosis in different organs, the results of protein and gene therapy may be inconsistent. To develop safe and effective methods to stimulate post-fibrosis lymphangiogenesis, much effort has been made in the continuous release of protein or gene delivery using adeno-associated viral (AAV) vectors (Table 2).

Currently, VEGFC/VEGFR3 is the most important lymphangiogenesis factor and has been studied often for the treatment of fibrosis. The application of soluble VEGFR3-Fc fusion protein in the mouse UUO model blocked the binding of VEGF C/VEGFD to VEGFR3, inhibited lymphangiogenesis and alleviated renal inflammation and fibrosis [5]. However, Hasegawa et al. also applied recombinant human VEGFC in the UUO model to ameliorate inflammation and fibrosis in the renal interstitium by inducing lymphangiogenesis [81]. They verified the effect of lymphangiogenesis on fibrosis by inhibiting lymphangiogenesis and promoting lymphangiogenesis. During the observation period of 1 week, Pei et al. injected VEGFR3 blockers into the UUO model once every 3 days, and Hasegawa et al. administered exogenous VEGFC to the UUO model once a day for 2 weeks. The difference in therapeutic effect may be due to the short observation time and the incomplete dose of lymphangiogenesis blocking. The long observation period and sufficient lymphatic vessels caused by sufficient VEGFC alleviated fibrosis in the same model. Similar to Hasegawa, in a rat model of myocardial fibrosis, targeted delivery of alginate albumin particles to the VEGFR3 selective design protein VEGFC (C152S) resulted in accelerated cardiac lymphangiogenesis in a dose-dependent manner, alleviated postmyocardial infarction inflammation and fibrosis, and improved cardiac edema and myocardial function [14]. However, there are also findings suggesting that intervention of lymphangiogenesis by VEGFC/VEGFR3 does not improve organ fibrosis. Yazdani et al. administered a neutralizing antibody for 6 consecutive weeks against anti-VEGFR3 in 6-week mouse models of albuminuria and blocked lymphangiogenesis but did not produce a therapeutic effect on fibrosis or inflammation [85]. In peritoneal fibrosis, adenovirus-expressing soluble VEGFR-3 (Adeno-sVEGFR-3) blocked lymphangiogenesis but had no significant effect on fibrosis, inflammation or neovascularization [23]. There may be differences in the efficacy of lymphatic intervention in different fibrosis models.

In addition to VEGFC/VEGFR3, macrophages are also the core factors of lymphangiogenesis, which can be regulated by macrophages through transdifferentiation into LECs and secretion of VEGFC. Therefore, the intervention of macrophages is also a potential therapeutic mode for fibrosis. Reinfusion of M2b macrophages into a rat myocardial infarction model promoted lymphangiogenesis, alleviated cardiac fibrosis, and improved cardiac function [86]. However, Lee et al. applied the macrophage scavenger clodronate to clear 92% of macrophages, reduce lymphangiogenesis by 85%, and alleviate fibrosis in a UUO model [84]. Similarly, Zhang et al. reached the same conclusion in the same model by Clodronate [8]. In contrast, Yazdani et al. inhibited macrophage inflow and partially decreased myofibroblast expression with chlorophosphonate liposomes but did not significantly prevent the development of lymphangiogenesis, fibrosis or albuminuria [85].

LYVE1 is also an important target for lymphangiogenesis and an important receptor for LECs, whose ligand is hyaluronic acid (HA) or fibroblast growth factor 2 (FGF2) [122]. Applying soluble LYVE1-Fc (sLYVE1-FC) blocked lymphangiogenesis in the UUO model and reduced inflammatory responses and fibrosis [5]. Although Vieira et al. did not target LYVE1 therapy, a similar effective conclusion was obtained in mouse cardiac fibrosis and LYVE1-dependent lymphangiogenesis to transport infiltrating immune cells to drainage mediastinal lymph nodes, thereby reducing myocardial inflammation and fibrosis. LYVE1 gene knockout prevented white blood cells from docking and transporting through the lymphatic endothelium. This leads to increased chronic inflammation and long-term deterioration of heart function [88]. Possibly due to organ differences, focusing on LYVE1 lymphangiogenesis has yielded different results, and further studies are needed to elucidate the underlying mechanism.

Overall, the mechanisms of macrophage-mediated, VEGFC/VEGFR3-mediated, and LYVE1-mediated lymphangiogenesis are well established according to all preclinical therapeutic models but differ in the outcomes of intervention in organ fibrosis. Zhang et al. [8] and Lee et al. [84] cleared macrophages in the UUO model, and Wang et al. [86] infused exogenous macrophages in the cardiac fibrosis model. All three experiments obtained a positive correlation between macrophages and lymphangiogenesis, but differences in fibrosis treatment results appeared, possibly related to the different organ fibrosis models. Notably, chlorophosphonate not only eliminates macrophages but also includes T cells and other cells [8]. However, Wang et al. transfused M2b subtype macrophages, and chlorophosphonate removed all subtype macrophages, which further complicated the intervention results and provided the possibility of different fibrotic results [86]. Hasegawa et al. [81] in a UUO model and Henri et al. [14] and Vieira et al. [88] in a myocardial infarction model promoted lymphangiogenesis by exogenous VEGFC, alleviated fibrosis and improved organ function, respectively. However, Yazdani et al. [85] in a rat model of albuminuria and Pei et al. [5] in a UUO model prevented lymphangiogenesis by inhibiting VEGFR3 or VEGFC/VEFDR3 binding. The former did not affect the fibrosis process, while the latter slowed fibrosis by inhibiting lymphangiogenesis, which was contrary to the results obtained by interfering with macrophage-mediated lymphangiogenesis. The former may be due to differences in models, while the latter may be because VEGFR3 is a plurality receptor of VEGF, not only binding with VEGFC.

Targeting the synergy between lymphangiogenesis and the inflammatory infiltrate is beneficial for promoting immune regulation and fibrosis recovery. Although gene therapy is promising for the future treatment of disease, treatments with local injection and intravascular infusion are challenging. In terms of delivery, viral vectors have disadvantages of infectivity, transgene ability, and expression stability. Tissue-specific adeno-associated viruses have relatively high transgenic time, but adenoviruses and plasmids have only short-term effects. Furthermore, neutralizing antibodies may limit the therapeutic effect and lead to a harmful immune response.

## 8. Conclusions and Views

Overall, current studies have clarified that lymphangiogenesis is associated with fibrotic diseases in multiple organs. First, based on the differences in disease models, the role of lymphangiogenesis in different fibrotic diseases is different. In terms of benefits, lymphatic vessels can remove locally infiltrating immune cells and slow the inflammatory response to relieve fibrosis. In contrast, distant circulating inflammatory cells are recruited to enhance inflammation and then worsen fibrosis. In addition, LECs may differentiate into myofibroblasts to exert a profibrotic effect. Second, the source of lymphangiogenesis or LECs in organ fibrosis consists of local LEC budding, proliferation of LEC progenitors and differentiation of macrophages. It is noteworthy that macrophage transdifferentiation may account for only a small fraction of lymphangiogenesis. Third, the VEGFC/VEGFR3 axis promotes lymphangiogenesis, and multiple studies have shown that macrophages can autocrine VEGFC to promote lymphangiogenesis in a VEGFC/VEGFR3-independent manner or act to transdifferentiate into LECs to participate in lymphangiogenesis. TGFβ inhibits lymphangiogenesis in multiple ways, and HA/LYVE1 promotes lymphangiogenesis. VEGFC/VEGFR3, TGFβ, HA/LYVE1 and other net factors, through the network form of complex interaction, jointly affect fibrosis-related lymphangiogenesis. Fourth, exogenous VEGFC, soluble VEGFR3 and LYVE1 interfered with lymphangiogenesis and improved organ fibrosis in a preclinical model, suggesting that intervention in lymphangiogenesis is a promising fibrosis treatment strategy.

## 9. Prospectives

Lymphangiogenesis plays an important role in improving the remission of inflammation and fibrosis as well as improving organ dysfunction, and has the potential to be one of the main therapeutic strategies for fibrotic diseases. However, several questions need to be addressed in future studies regarding the positive and negative effects of lymphangiogenesis in the pathological process of organ fibrosis: Is the effect of lymphangiogenesis on fibrosis in multiple organs good or bad? How does lymphangiogenesis affect the progression or remission of fibrosis? What is the main mechanism of lymphangiogenesis? All of these directly affect the strategies and outcomes of treating fibrosis through lymphangiogenesis intervention. Secondly, an increasing number of experimental studies using gene knockout or transgenic mice show that up- or down-regulation of these factors can alter the course of fibrosis, and therapeutic strategies targeting lymphangiogenesis may be promising. However, the validity of these treatments should be confirmed in additional animal models and for longer time periods before considering clinical trials. Furthermore, potential adverse effects should be closely watched, as some lymphangiogenic factors have been shown to be protective in some cases but simultaneously promote other disease processes. Finally, because the factors that regulate lymphangiogenesis have different pathophysiological roles in different organs and tissues, specific drug delivery systems are needed to deliver these factors to the target organs for precise treatment.

## Figures and Tables

**Figure 1 ijms-24-14123-f001:**
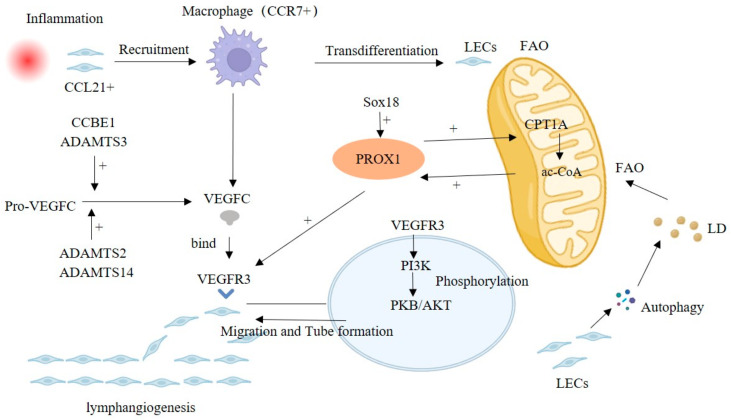
Classical VEGFC/VEGFR3-mediated lymphangiogenesis pathway, including FAO, macrophages and crosstalk signaling. Abbreviation: FAO, Fatty acid β-oxidation; LD, Lipid droplet.

**Figure 2 ijms-24-14123-f002:**
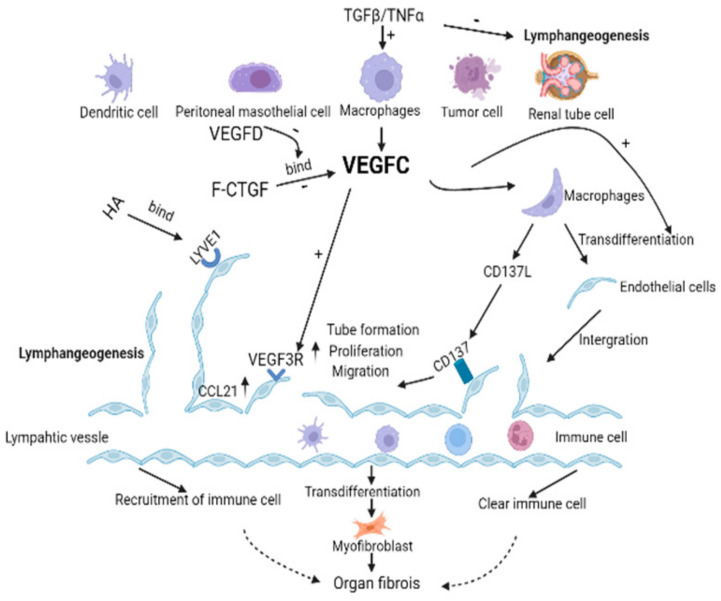
VEGFC promotes lymphangiogenesis in the pathological environment of organ fibrosis. VEGFC can bind to the receptor VEGFR3 of LECs to promote the proliferation and migration of LECs as well as tube formation. In addition, the literature also reported that HA binds to the receptor LYVE 1 on LECs, and CD137L binding to the receptor CD137 on LECs could also promote the biological behavior of LECs. Moreover, VEGFC promotes lymphangiogenesis in a manner that promotes macrophages partially transdifferentiate into LECs. Lymphatic vessels are the transport channel of immune cells, changing the course of fibrosis by introducing or removing immune cells in different disease backgrounds. LECs can also directly differentiate into myofibroblasts to promote fibrosis. TGF- β and TGF- α can promote the production of VEGFC in tissues and cells in the context of multiple fibrosis, but it can directly inhibit lymphangiogenesis. Full-length connective tissue growth factor (F-CTGF) binding to VEGFC inhibited lymphangiogenesis, but this effect was abolished by the VEGFD. In conclusion, VEGFC promotes lymphangiogenesis and mediates the course of the fibrotic disease. Solid arrow means means facts supported by the literature, and dashed arrow indicate the currently considered differential effect of immune cell infiltration on organ fibrosis.

**Table 2 ijms-24-14123-t002:** Organ fibrosis therapy targeting lymphangiogenesis.

Model	Lymphangiogenesis Intervention	Lymphangiogenesis Change	Therapy Results	Reference
UUO	Clodronate	down	Alleviation the fibrosis	[84]
UUO	Clodronate	down	Alleviating the fibrosis	[8]
UUO	recombinant human VEGFC	up	Alleviating the fibrosis	[81]
UUO	Soluble VEGFR3-Fc fusion protein (sVEGFR3-FC), soluble LYVE-1-Fc (sLYVE-1-FC)	down	Alleviating the fibrosis	[5]
MI	VEGFR3-selective designer protein VEGFC-C152S	up	Alleviating the fibrosis and inflammation, dysfunction of heart	[14]
MI	VEGFC(C156S)	up	Alleviated the inflammation of myocardial infarction	[88]
MF	M2b macrophage transplantation	up	Reducing fibrosis, improving the left ventricular ejection fraction	[86]
Proteinuria	Anti-VEGFR3 antibody	down	No	[85]

UUO, unilateral ureteral obstruction. MI, myocardial infarction. MF, myocardial fibrosis.

## Data Availability

Not applicable.

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
