# Peer review of "Angiogenesis—An Emerging Role in Organ Fibrosis"

_ijms, 2023, doi:10.3390/ijms241814123_

Round 1

Reviewer 1 Report (New Reviewer)

This review is about an interesting problem related to the role of lymphangiogenesis in fibrosis development. The problem is difficult with opposite results, and this review allows to organize and clarify the data.

To improve the quality of the paper, I propose to:

- Add two paragraphs about the mechanisms of lymphangiogenesis and the involved signaling pathways. There is currently a paragraph "Lymphangiogenesis pathway" but it only describes the binding of VEGF-C to VEGFR3. We need to get more about the downstream signaling pathways.

- The paper miss iconography. I propose authors add two figures concerning these two points.

- The quality of the paper could be improved with a proofreading by an appropriate organism.

The english is quite good but I think it could be of interest to improve the quality of the paper with a proofreading by an appropriate organism.

Author Response

Dear Professor,

Thank you very much for your valuable comments on our manuscript during your busy schedule. We responded item by item as requested.

1.Add two paragraphs about the mechanisms of lymphangiogenesis and the involved signaling pathways. There is currently a paragraph "Lymphangiogenesis pathway" but it only describes the binding of VEGF-C to VEGFR3. We need to get more about the downstream signaling pathways.

Response: we have added signaling pathways in the script (line 79-124 and marked red)

2.The paper miss iconography. I propose authors add two figures concerning these two points.

Response: In the manuscript, we made two figures in total.

Figure1 depicts the pathway of lymphangiogenesis, and Figure2 describes the lymphangiogenic pathway associated with organ fibrosis. We look forward to the professor can give more valuable suggestions and improve our manuscript, because we have learn a lot from the process.

3.The quality of the paper could be improved with a proofreading by an appropriate organism.

Response: We have asked scholars in the research field to make revisions. If there is any revision, please communicate with us in time. Thank you for your hard review of our manuscript.

Reviewer 2 Report (Previous Reviewer 1)

The revised manuscript is much improved and more informative. The authors have made significant changes and addressed all my concerns. I feel that in the present form this manuscript provides new insights in the pathophysiology of lymphoangiogenesis.  I have no further comments.

English is OK.

Author Response

Dear Professor,

Thank you very much for your valuable comments on our manuscript during your busy schedule. We responded item by item as requested.

1.The revised manuscript is much improved and more informative. The authors have made significant changes and addressed all my concerns. I feel that in the present form this manuscript provides new insights in the pathophysiology of lymphoangiogenesis.  I have no further comments.

Response:Thank you for your suggestions to improve our manuscript.

This manuscript is a resubmission of an earlier submission. The following is a list of the peer review reports and author responses from that submission.

Round 1

Reviewer 1 Report

The authors review the potential relevance and link of lymphatic vessels in the patho-physiology of fibrosis in organs (kidney, liver, heart and some additional organs) other than tumors/cancers. To highlight the importance the authors describe the well established basics of lymphatic endothelial characteristics and the major mechanism(s) driving their growth i.e. VEGF/VEGFR3 and TGF-beta.  I have a few suggestions:

1: Since most of the work reviewed is from work using dermal lymphatic endothelial cells or cancer tissue lymphatic ECs, the authors may want to emphasize more on the differences in lymphangiogenesis in various organs. 2: The authors discuss two major mechanism(s) i.e. VEGF and TGF-beta. However, it is well known that other small molecules i.e. microRNAs also regulate LEC growth and may be driving fibrosis by modulating both VEGF and TGF-beta. In this context the authors should review and integrate the recent work done by Giovanna Azzarito (Azzarito et al,  IJMS 2022; 23 (13), T192 ; Azzarito et al. Cells 2023, 12(3) 389 https://doi.org/.10.3390/cells 12030389) from Prof. Dubey's  lab in Switzerland.  Additionally, the authors should look into the microarray data from the above articles to see how fibrosis associated genes are being regulated. Providing insights and postulating novel directions would help the readers to focus on the novel fibrosis associated mechanisms that may be triggered by or in LECs.  I suggest that the authors 

3: The table mentioned in the review was not provided.

4: The authors should provide some cartoons/figures to make their review more attractive and useful for the audience.

The english is OK

Author Response

Dear reviewer,

Thank you for your suggestions for our manuscript. Our responses to the questions raised are as follows.

1: Since most of the work reviewed is from work using dermal lymphatic endothelial cells or cancer tissue lymphatic ECs, the authors may want to emphasize more on the differences in lymphangiogenesis in various organs. 2: The authors discuss two major mechanism(s) i.e. VEGF and TGF-beta. However, it is well known that other small molecules i.e. microRNAs also regulate LEC growth and may be driving fibrosis by modulating both VEGF and TGF-beta. In this context the authors should review and integrate the recent work done by Giovanna Azzarito (Azzarito et al,  IJMS 2022; 23 (13), T192 ; Azzarito et al. Cells 2023, 12(3) 389 https://doi.org/.10.3390/cells 12030389) from Prof. Dubey's  lab in Switzerland.  Additionally, the authors should look into the microarray data from the above articles to see how fibrosis associated genes are being regulated. Providing insights and postulating novel directions would help the readers to focus on the novel fibrosis associated mechanisms that may be triggered by or in LECs.  I suggest that the authors 

Response:We added some research reports on miRNAs and lymphatic vessels, and elaborated them in detail. (line 385-402 and marked red)

2: The table mentioned in the review was not provided.

Response:we have added the table in the revised manuscript.

3: The authors should provide some cartoons/figures to make their review more attractive and useful for the audience.

Response:we have added the Figure in the revised manuscript.

Reviewer 2 Report

Wang and colleagues comprehensively summarised the current studies on fibrosis and lymphangiogenesis, focusing on its potential therapeutic potential for these diseases. While the middle part was well written, the final perspective section requires extensive rewriting. Many claims are quite vague without any reference as well. Some comments below:

Line 124 – Is liver the largest lymph producing organ instead of lymphatic? Lymphatic and lymph are different. This applies to line 139 as well.

Throughout the review, authors tend to go in and out topic, making it extremely difficult to follow the theme. For example, in the heart section, the authors suddenly talk about lymphangiogenesis and differentiation between line 162-166. Then go on to mention that sources for LECs may be non-venous in the heart. The basic biology on lymphatic development should be covered in the initial introductory section.

Also, in the liver section, the author talks about anti-lymphangiogenic drugs and their clinical trials, seemingly out of no where. The author also mentions anti-lymphangiogenic therapy in the previous kidney section but without going into detail of its clinical outcome like in the liver.

The review would be significantly improved if each section in “The role of lymphatic vessels in tissue and organs” stays on topic.

Heart – there are significant literature on the role of heart lymphatics in regeneration in zebrafish and mice – these should be discussed.

Other organs with significant study on lymphatic biology incudes the gut and brain. These should be mentioned in the Other organs section.

Line 452 -lymphangiogenesis instead of lymphangiotropin?

Lines 450 onwards – lack of any referencing.

The perspective section is very poorly written. Very difficult to understand and require extensive proof reading. Many vague statements with absolutely no backing or reference. For example, “A proportion of the teams conducted related studies but received only lim- 508

ited evidence.” Does not mean much as it is unclear which proportion of team conducted what study. This section is the most important as it highlights the future direction and thoughts of the author and it concludes the whole review.

The role of lymphangiogenesis in fibrosis is still unclear and as author suggested, context dependent. A table showing each study and whether therapeutic intervention (either anti or pro lymphangiogenesis or macrophage ablation) alleviated fibrosis in various models should be made. This table would be a powerful reference to quickly glance the outcomes of each interventions in different fibrosis models.

I did not see any Figure or the 2 tables in the submitted manuscript. Please provide these for revision.

Extensive proof reading/rewriting needed as suggested. 

Author Response

Dear reviewer,

Thank you for your suggestions for our manuscript. Part line changes due to the modification of the manuscript, and we make some marks. Our responses to the questions raised are as follows.

1.Line 124 – Is liver the largest lymph producing organ instead of lymphatic? Lymphatic and lymph are different. This applies to line 139 as well.

Response:Thank you for raising this issue. We have already made the modification. (line 113and 128, marked red)

2.Throughout the review, authors tend to go in and out topic, making it extremely difficult to follow the theme. For example, in the heart section, the authors suddenly talk about lymphangiogenesis and differentiation between line 162-166. Then go on to mention that sources for LECs may be non-venous in the heart. The basic biology on lymphatic development should be covered in the initial introductory section.

Response: Thank you for your pertinent comments and we have revised them. We have comprehensively modified the heart and kidney sections, we have removed the development and distribution of lymphatic vessels in organ (line 95-111 and line 145-164, marked red)

3.Also, in the liver section, the author talks about anti-lymphangiogenic drugs and their clinical trials, seemingly out of no where. The author also mentions anti-lymphangiogenic therapy in the previous kidney section but without going into detail of its clinical outcome like in the liver.

Response: Thank you put forward this problem, we heard the clinical trials about VGX-100 in the liver section on a lecture and got clinical trial number (NCT01514123), but we ignored in no website retrieved (NCBI) and can’t be cited. Therefore, we have removed this part from the manuscript (previous line 146-148).

This clinical trial is actually available on the general website, and it turns out 36% of patients with advanced tumors had an excellent response to VGX-100,but phase II trials have not yet begun. (https://xueshu.baidu.com/usercenter/paper/show?paperid=e79ed84cd9918975ce97f84388d97d79)

In the kidney section, we have added the references in the marked red part.

4.Heart – there are significant literature on the role of heart lymphatics in regeneration in zebrafish and mice – these should be discussed.

Response: We add the study of heart lymphangiogenesis in regeneration in zebrafish and mice. (line 157-163 and marked red plus bold)

5.Other organs with significant study on lymphatic biology incudes the gut and brain. These should be mentioned in the Other organs section.

Response: We have added lymphatic biology of gut and brain in the “other organs” section (line 166-178 and marked red)

6.Line 452 -lymphangiogenesis instead of lymphangiotropin?

Response: Yes, we have modified it line 464 (previous 452) and marked red due to increasing content in the context.

7.Lines 450 onwards – lack of any referencing.

Response: we have added the references (line 463-482).

8.The perspective section is very poorly written. Very difficult to understand and require extensive proof reading. Many vague statements with absolutely no backing or reference. For example, “A proportion of the teams conducted related studies but received only lim- 508

ited evidence.” Does not mean much as it is unclear which proportion of team conducted what study. This section is the most important as it highlights the future direction and thoughts of the author and it concludes the whole review.

Response: Thank you for your careful review of our manuscript. We have carefully rewritten the perspective section and learned a lot again. If we need to revise it, we are very happy that you can make better suggestions to improve our manuscript.

9.The role of lymphangiogenesis in fibrosis is still unclear and as author suggested, context dependent. A table showing each study and whether therapeutic intervention (either anti or pro lymphangiogenesis or macrophage ablation) alleviated fibrosis in various models should be made. This table would be a powerful reference to quickly glance the outcomes of each interventions in different fibrosis models.

Response: We have added the table in the context, Tableâ… shows effect of lymphangiogenesis on organ fibrosis and the underlying mechanisms. Tableâ…¡shows effect therapeutic intervention on organ fibrosis.

10.I did not see any Figure or the 2 tables in the submitted manuscript. Please provide these for revision.

Response: we have added the table and figure in the manuscript.